# INFERRING DNN-BRAIN ALIGNMENT USING REPRESENTATIONAL SIMILARITY ANALYSES CAN BE PROBLEMATIC

**Marin Dujmović & Jeffrey S Bowers**
School of Psychological Science
University of Bristol
Bristol, UK
{marin.dujmovic,j.bowers}@bristol.ac.uk

**Federico Adolfi**
Ernst Strüngmann Institute for Neuroscience
in Cooperation with Max-Planck Society
Frankfurt, Germany
fedeadolfi@gmail.com

**Gaurav Malhotra**
University at Albany, SUNY
NY 12222, USA
gmalhotra@albany.edu

## ABSTRACT

Representational Similarity Analysis (RSA) has been used to compare representations across individuals, species, and computational models. Here we focus on comparisons made between the activity of hidden units in Deep Neural Networks (DNNs) trained to classify objects and neural activations in visual cortex. In this context, DNNs that obtain high RSA scores are often described as good models of biological vision, a conclusion at odds with the failure of DNNs to account for the results of most vision experiments reported in psychology. How can these two sets of findings be reconciled? Here, we demonstrate that high RSA scores can easily be obtained between two systems that classify objects in qualitatively different ways when second-order confounds are present in image datasets. We argue that these confounds likely exist in the datasets used in current and past research. If RSA is going to be used as a tool to study DNN-human alignment, it will be necessary to experimentally manipulate images in ways that remove these confounds. We hope our simulations motivate researchers to reexamine the conclusions they draw from past research and focus more on RSA studies that manipulate images in theoretically motivated ways.

## 1 INTRODUCTION

How do other animals see the world? Do different species represent the world in a similar manner? How do the internal representations of AI systems compare with humans and animals? The traditional scientific method of probing internal representations of humans and animals (popular in both psychology and neuroscience) relates them to properties of the external world. By moving a line across the visual field of a cat, Hubel & Wiesel (1959) found out that neurons in the visual cortex represent edges moving in specific directions. In another Nobel-prize winning work, O'Keefe (1976) and Hafting et al. (2005) discovered that neurons in the hippocampus and entorhinal cortex represent the location of an animal in the external world. Despite these successes it has proved difficult to relate internal representations in real-world settings, where the visual system processes complex, high-dimensional sensory inputs.

These challenges have contributed to the recent excitement around Representation Similarity Analysis (RSA), which is a multi-voxel pattern analysis method specifically designed to compare representations between different systems processing high-dimensional data. RSA usually takes patterns of activity from two systems and computes how the distances between activations in one system correlate with the distances between corresponding activations in the second system (see Figure A1). Rather than compare each pattern of activation in the first system directly to the corresponding pat-

tern of activation in the second system, it computes representational distance matrices (RDMs), a *second-order* measure of similarity that compares systems based on the relative distances between neural response patterns. This arrangement of neural response patterns in a representational space has been called a system's *representational geometry* (Kriegeskorte & Kievit, 2013). The advantage of looking at representational geometries is that one no longer needs to match the architecture of two systems, or even the feature space of the two activity patterns. One could compare, for example, fMRI signals with single cell recordings, EEG traces with behavioural data, or vectors in a computer algorithm with spiking activity of neurons (Kriegeskorte et al., 2008a). RSA is now ubiquitous in computational psychology and neuroscience and has been applied to compare object representations in humans and primates (Kriegeskorte et al., 2008b), representations of visual scenes by different individuals (Haxby et al., 2011; O'Hearn et al., 2020), representations of visual scenes in different parts of the brain (Mack et al., 2013), to study specific processes such as cognitive control (Freund et al., 2021) or the dynamics of object processing (Kaneshiro et al., 2015), and most recently, to relate neuronal activations in human (and primate) visual cortex with activations of units in Deep Neural Networks (Yamins et al., 2014; Khaligh-Razavi & Kriegeskorte, 2014; Kietzmann et al., 2019; Cichy et al., 2016; Kiat et al., 2022).

However, this flexibility in the application of RSA comes at the price of increased ambiguity in the inferences one can draw from this analysis. Since RSA score is a second-order summary statistic (it looks at similarity of similarities), it loses information about how inputs are encoded by a system. Most relevant to present concerns, RSA is ambiguous with respect to the *stimulus features* that drive the observed similarity between two systems (Haxby et al., 2014; Diedrichsen & Kriegeskorte, 2017). That is, two systems that operate on completely different stimulus features can still have highly correlated representational geometries. Nevertheless, many researchers continue to infer that two systems are mechanistically similar based on RSA-scores, perhaps on the assumption that this is unlikely to occur in practice. Here we carry out two simulation studies showing that this problem needs to be taken seriously when drawing inferences regarding DNN-brain alignment on the basis of RSAs. See Figure A2 in Appendix for an illustration of how two different systems encoding different features can nevertheless produce high RSA scores.

## 2 STUDY 1: COMPARING DIFFERENT CNNs WITH RSA

We trained four different DNNs – $\{\Phi_1, \Phi_2, \Phi_3, \Phi_4\}$ – to classify input images. $\Phi_1$ was trained on an unperturbed CIFAR-10 dataset, while the remaining three were trained on modified versions of CIFAR-10 that contained a confound, namely, single pixels at locations diagnostic of the category. For $\Phi_2$, the locations of these diagnostic pixels were chosen such that they were positively correlated to the corresponding representational distances between classes in $\Phi_1$. For $\Phi_3$ and $\Phi_4$, the location of the confound was uncorrelated and negatively correlated with $\Phi_1$'s RDM, respectively. So $\Phi_2$–$\Phi_4$ all share a first order confound with $\Phi_1$ (a pixel location diagnostic of object category), but $\Phi_2$ shares an additional second order confound that might inflate the RSA score. Our hypothesis is that this second order confound will result in $\Phi_1$ & $\Phi_2$ learning similar representational geometries despite using qualitatively different features for classification, leading to a higher $\Phi_1$–$\Phi_2$ RSA scores compared to $\Phi_1$–$\Phi_3$ or $\Phi_1$–$\Phi_4$. For illustration of how we added the pixel confounds, see Figure A3 in Appendix.

### 2.1 DESIGN AND PROCEDURE

For all DNNs we used A `VGG-16` deep convolutional neural network (Simonyan & Zisserman, 2014) pre-trained on the `ImageNet` dataset of naturalistic images. $\Phi_1$ was trained to classify stimuli from the `CIFAR-10` dataset (Krizhevsky & Hinton, 2009). The `CIFAR-10` dataset includes 10 categories with 5000 training, and 1000 test images per category. The network was fine-tuned on `CIFAR-10` by replacing the classifier so that the final fully-connected layer reflected the correct number of target classes in `CIFAR-10` (10 for `CIFAR-10` as opposed to 1000 for `ImageNet`). Images were rescaled to a size of $224 \times 224$px and then the model learnt to minimise the cross-entropy error using the RMSprop optimizer with a mini-batch size of 64, learning rate of $10^{-5}$, and momentum of 0.9. All models were trained for 10 epochs, which were sufficient for convergence across all datasets.

We then computed the RDM for $\Phi_1$. One hundred random images from the test set for each category were sampled as input for the network and activations at the final convolutional layer extracted using the `THINGSVision` Python toolkit (Muttenthaler & Hebart, 2021). The same toolkit was used to generate a representational dissimilarity matrix (RDM) from the pattern of activations using `1-Pearson's r` as the distance metric. The RDM was then averaged by calculating the median distance between each instance of one category with each instance of the others (e.g., the median distance between `Airplane` and `Ship` was the median of all pair-wise distances between activity patterns for airplane and ship stimuli). This resulted in a $10 \times 10$, category-level, RDM which reflected median between-category distances.

Next, the three modified versions of the `CIFAR-10` datasets were created for the 'Positive', 'Uncorrelated' and 'Negative' conditions, respectively. In the 'Positive' condition the distances between pixel placements was positively correlated with the RDM of $\Phi_1$. We achieved this by using an iterative algorithm that sampled pixel placements at random, calculated an RDM based on distances between the pixel placements and computed an RSA score (Spearman correlation) with the target RDM. Placements with a score above $0.70$ were retained and further optimized (using small perturbations) to achieve an RSA-score over $0.90$. The same procedure was also used to determine placements in the Uncorrelated (optimizing for a score close to 0) and Negatively correlated (optimizing for a negative score) conditions. Finally, datasets were created using 10 different placements in each of the three conditions. Networks were trained for classification on these modified `CIFAR-10` datasets in the same manner as the `VGG-16` network trained on the unperturbed version of the dataset (See Step 1 above).

Finally, once all four DNNs had been trained, we tested them on withheld examples of both unperturbed and single-pixel confound images, in a 4 x 2 design. We measured classification performance of each type of network as well as RSA-scores between pairs of networks. That is, we measured how the representation geometries compared between these networks trained to select different features of the inputs. All image-level RDMs were calculated using $1 - r$ as the distance measure and RSA-scores were computed as the Spearman rank correlation between RDMs. In each case, these RDMs were obtained for a set of 50 images (giving a $50 \times 50$ RDM), with 5 images from each of the 10 categories of images in the `CIFAR-10` dataset. These 5 images were obtained by computing the correlation of average activation elicited by an image with the average activation for the entire category. We then selected the 5 images eliciting the highest correlation with average activation for each category. In this manner we obtained the most prototypical images for each category.

To determine the highest-possible RSA-score given the variation in RDMs within each network, we determined a noise ceiling. After appropriate transformations (see Nili et al., 2014), an average RDM of the target system (in this case networks trained on naturalistic images) is computed. The upper bound of the noise ceiling is given by the average RSA score between that RDM and each network trained on naturalistic images. The lower bound is calculated by employing a leave-one-out procedure. RSA scores are computed between each network trained on naturalistic images and the average RDM of the remaining networks. The average of those RSA scores represents a lower bound of the noise ceiling. All DNN simulations were carried out using the `Pytorch` framework (Paszke et al., 2019). The model implementations were downloaded from the `torchvision` library. Networks trained on unperturbed as well as modified datasets were pre-trained on `ImageNet` and their pre-trained weights were downloaded from `torchvision.models` subpackage.

## 2.2 RESULTS

Classification performance (Figure 1, left) was analyzed by conducting a 4 (network) by 2 (dataset with/without confound) ANOVA. Results revealed a significant interaction effect ($F(3, 36) = 12256.10, p < .001, \eta_p^2 = .99$). Post-hoc analysis (Tukey HSD) showed that the network $\Phi_1$, trained on unperturbed images, learned to classify these images and ignored the diagnostic pixel – that is, it's performance was identical for the unperturbed and modified images ($p = .990$). In contrast, networks $\Phi_2$ (positive), $\Phi_3$ (uncorrelated) and $\Phi_4$(negative) failed to classify unperturbed images (performance was near chance) but learned to perfectly classify the modified images (all $p < .001$), showing that these networks develop qualitatively different representations compared to normally trained networks.

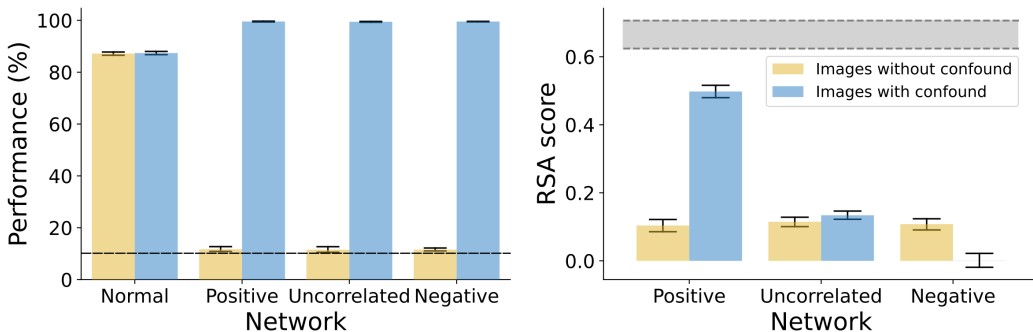

Figure 1: Study 1 result. *Left:* Performance of normally trained networks did not depend on whether classification was done on unperturbed `CIFAR-10` images or images with a single pixel confound (error bars represent 95% CI, the dashed line represents chance performance). All three networks trained on datasets with confounds could perfectly categorise the test images when they contained the confound they were trained on (blue bars), but failed to achieve above-chance performance if the predictive pixel was not present (yellow bars). *Right:* The RSA score between the network trained on the unperturbed dataset and each of the networks trained on datasets with confounds. The three networks showed similar scores when tested on images without confounds, but vastly different RSA scores when tested on images with confounds. Networks in the Positive condition showed near ceiling scores (the shaded area represents noise ceiling) while networks in the Uncorrelated and Negative conditions showed much lower RSA.

Next we computed pairwise RSA scores between the representations at the last convolution layer of $\Phi_1$ and each of $\Phi_2, \Phi_3$ and $\Phi_4$ (Figure 1, right). The data was then analyzed by conducting a 3 (network) by 2 (dataset) ANOVA which revealed a significant interaction effect ($F(2, 297) = 289.27, p < .001, \eta_p^2 = .66$). When presented unperturbed test images, the $\Phi_2, \Phi_3$ and $\Phi_4$ networks all showed low (and equal; all $p > .954$) RSA scores with the normally trained $\Phi_1$ network. However, when networks were presented with test images that included the predictive pixels, RSA varied depending on the geometry of pixel locations in the input space. When the geometry of pixel locations was positively correlated to the normally trained network, RSA scores approached ceiling (i.e., comparable to RSA scores between two normally trained networks). Networks trained on uncorrelated and negatively correlated pixel placements scored much lower (all $p < .001$).

## 3 STUDY 2: COMPARING CNNs AND HUMAN IT CORTEX WITH RSA

In Study 1 we showed that CNNs that classify objects using qualitatively different visual features can nevertheless learn similar representational geometries due to second order confounds. However, it might be argued that this pattern of results is easier to obtain between two CNNs than between a CNN and a brain. Furthermore, our use of the RDM from $\Phi_1$ to place the pixel confounds might be questioned. Perhaps the high $\Phi_1 - \Phi_2$ RSA depended on our carefully crafted confound, and it is unlikely that similar confounds exist in the images used in past research. Accordingly, we carried out a second study that assessed RSAs between DNNs and human brain data, and we placed the pixel confound in images without recourse to the RDM.

We selected a popular dataset used for comparing representational geometries in humans, macaques and deep learning models (Khaligh-Razavi & Kriegeskorte, 2014; Kriegeskorte, 2009). This dataset consists of six categories which can be organised into a hierarchical structure shown in Figure 2. Kriegeskorte et al. (2008b) showed a striking match in RDMs for response patterns elicited by these stimuli in human and macaque IT, and subsequently, Khaligh-Razavi & Kriegeskorte (2014) reported a high RSA bewtween a CNN and human IT cortex using this dataset.

The BOLD signal from humans viewing these stimuli constituted $\Phi_1$. As in Study 1, we trained a network to classify unperturbed stimuli ($\Phi_2$) as well as images that contained a confound of a single predictive pixel. But instead of using an RDM to compute the location of a diagnostic pixel, we used the hierarchical categorical structure. $\Phi_3$ was trained on the first modified version of the dataset,

where the location of the pixel was based on the hierarchical structure of categories in Figure 2. One such configuration can be seen in Figure 2. $\Phi_4$ was trained on the second modified version of the dataset, where the predictive pixel was placed at a random location for each category (but, of course, at the same location for all images within each category). We call these conditions 'Hierarchical' and 'Random', respectively. The critical question is how the RSA betweem $\Phi_1 - \Phi_2$ (the condition previously explored in Khaligh-Razavi & Kriegeskorte, 2014) compared to the hierarchical $\Phi_1 - \Phi_3$ and random $\Phi_1 - \Phi_4$ conditions.

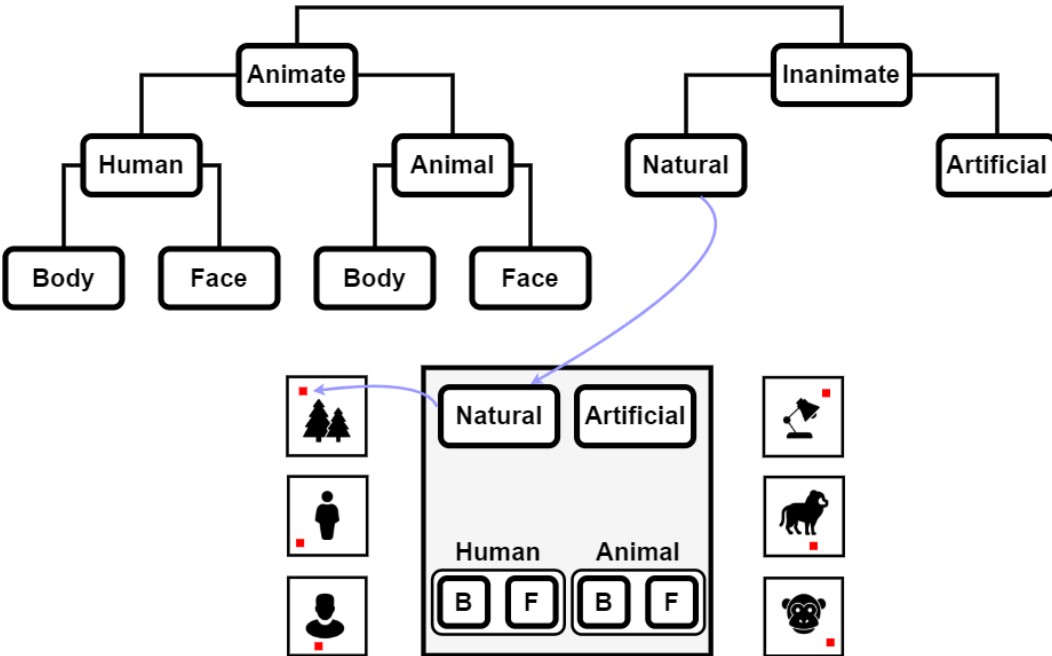

Figure 2: Exploiting intrinsic dataset hierarchy in order to place confounds. The top panel shows the hierarchical structure of categories in the dataset that was used to place the single pixel confounds. The example at the bottom (middle) shows one such hierarchical placement scheme where the pixels for Inanimate images were closer to the top of the canvas while Animate images were closer to the bottom. Within the Animate images, the pixels for Humans and Animals were placed at the left and right, respectively, and the pixels for bodies (B) and faces (F) were clustered as shown.

## 3.1 PROCEDURE

Rather than calculating pixel placements based on the human RDM, the hierarchical structure of the dataset was used to place the pixels manually. The dataset consists of 910 images from 6 categories: human bodies, human faces, animal bodies, animal faces, artificial inanimate objects and natural inanimate objects. These low-level categories can be organised into the hierarchical structure shown in Figure 2. Predictive pixels were manually placed so that the distance between pixels for Animate kinds were closer together than they were to Inanimate kinds and that faces were closer together than bodies. This can be done in many different ways, so we created five different datasets, with five possible arrangements of predictive pixels. Results in the Hieararchical condition (Figure 3) are averaged over these five datasets. Placements for the Random condition were done similarly, except that the locations were selected randomly. Networks (VGG-16) were then trained on a 6-way classification task in a similar manner to the previous study. Networks trained on the modified datasets (both Hierarchical and Random conditions) were not pre-trained on `ImageNet`. Models pre-trained on `ImageNet` were chosen for tuning with the unperturbed dataset recisely so that they attain good RSA with human neural activity. Contrary to this, networks trained on datasets with confounds were not pre-trained precisely to make our manipulation less likely to succeed in resulting with high RSA with human neural activity. In effect, the choices were made to provide a more difficult test for our hypothses.

The dataset used in this study consisted of $818$ training images and $92$ test images. Kriegeskorte et al. (2008b) used these images to obtain a $92 \times 92$ RDM to compare representations between human and macaque IT cortex. Here we computed a similar $92 \times 92$ RDM for networks trained in the Normal, Hierarchical and Random training conditions, which were then compared with the $92 \times 92$ RDM from human IT cortex to obtain RSA-scores for each condition. Like in Study 1 we also computed the performance of each DNN for test images with and without confounds. And again, as in Study 1, VGG-16 implementations were downloaded from the `torchvision` library. Networks trained on modified datasets were randomly initialised. For the pre-trained models, their pre-trained weights were downloaded from `torchvision.models` subpackage.

## 3.2 RESULTS

Results for this study are shown in Figure 3. Similarly to Study 1, analysis of performance resulted with a significant 3 (network) by 2 (dataset) interaction effect ($F(2, 42) = 407.61, p < .001, \eta_p^2 = .95$). Performance for networks that were trained on datasets with Hierarchically ($\Phi_3$) and Randomly placed pixels ($\Phi_4$) was much lower when presented with unperturbed images (both $p < .001$). At the same time, networks trained on naturalistic images ($\Phi_2$) achieved the same level of performance regardless of dataset ($p > .999$).

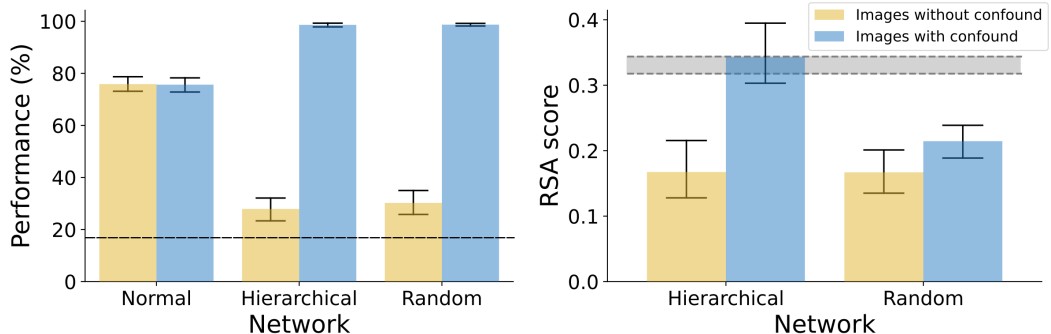

Figure 3: Study 2 results. *Left:* Performance of normally trained networks did not depend on whether the confound was present. Networks trained with the confound failed to classify stimuli without the confound (yellow bars) while achieving near perfect classification of stimuli with the confound present (blue bars, dashed line represents chance performance). *Right:* RSA with human IT activations reveals that, when the confound was present, the RSA-score for networks in the Hierarchical condition matched the RSA-score of normally trained network (gray band), while the RSA-score of the network in the Random condition was significantly lower. The grey band represents 95% CI for the RSA score between normally trained networks and human IT.

Further, we observed that there was no difference in RSA-scores with human IT ($\Phi_1$) between a network trained on Hierarchically placed pixels and a network trained on naturalistic images for datasets in which the confound was present ($t(28) = 0.46, p = .647$). However, when the pixel locations for each category were randomly chosen, the RSA-score decreased significantly. There was no difference between networks trained on Hierarchical and Random pixel placements when presented with unperturbed images. This was confirmed by a significant 2 (network) by 2 (dataset) interaction effect and follow-up tests ($F(1, 28) = 8.46, p = .007, \eta_p^2 = .23$). These results suggest that any confound in the dataset (including texture, colour or low-level visual information) that has distances governed by the hierarchical clustering structure of the data could underlie the observed similarity in representational geometries between CNNs and human IT. More generally, these results show how it is plausible that many confounds present in popular datasets may underlie the observed similarity in representational geometries between two systems. The error of inferring a similarity in mechanism based on a high RSA score is not just possible but also probable.

## 4 GENERAL DISCUSSION

In two studies, we showed that visual systems that rely on different visual features to identify objects can nevertheless obtain high RSA scores due to second order confounds in images. In Study 1 this was observed between two CNNs, and in Study 2, between a CNN and the human brain. Researchers have noted in the past that RSA is ambiguous with respect to the stimulus features that drive the observed similarity between two systems (e.g., Diedrichsen & Kriegeskorte, 2017), and nevertheless, high RSA scores are often taken as evidence that DNNs are promising models of human vision. Perhaps it is widely assumed that high RSA is unlikely between two systems that operate on different features. Our simulations provide the first empirical demonstration that this outcome is not only possible but also plausible given the high dimensionality and the hierarchical structure of images used in most RSA studies.

### 4.1 SIGNIFICANCE AND IMPLICATIONS

These results are critical for researchers making claims regarding the similarity of two systems using RSA. This includes research comparing information processing across species (Kriegeskorte et al., 2008b), across brain areas (Mack et al., 2013), between computational models (Raghu et al., 2021) and between artificial intelligence models and brains (Yamins et al., 2014; Khaligh-Razavi & Kriegeskorte, 2014; Kietzmann et al., 2019; Cichy et al., 2016; Kiat et al., 2022). Our findings are particularly relevant to an ongoing debate regarding the relation between DNNs and biological vision. Studies comparing DNNs to the visual cortex on the basis of RSA are often taken to provide evidence that DNNs and humans are well aligned (Cichy et al., 2016; Kiat et al., 2022; Mehrer et al., 2021; Tarigopula et al., 2023). By contrast, the failure of DNNs to account for a wide variety of key vision experiments reported in psychology has been taken as evidence that they are not well aligned (Bowers et al., 2023; Dujmović et al., 2020; Serre, 2019). How can these two sets of findings be reconciled?

Our suggestion is that the high RSAs may often reflect second order confounds present in images rather than good alignment. For example, most DNNs often classify objects on the basis of texture and local features whereas humans rely on global shape representations (Baker & Elder, 2022; Geirhos et al., 2018; Malhotra et al., 2022). Nevertheless, these same DNNs appear to be well aligned in terms of their RSA scores. One possible explanation for these two sets of findings is that texture is a second order confound with shape, such that the texture representational geometries in DNNs correlate with the shape representational geometries in humans. In which case, the high RSA scores should not be used to conclude that DNNs are a good model of human object recognition. Or alternatively, texture representations might be the most salient signals in both the DNNs and human studies, and high RSAs reflect an alignment in terms of how DNNs and humans encode texture. But current research does not allow us to distinguish between these two hypotheses, and even if DNNs and humans are aligned in how they represent texture and local features, it would be unwarranted to conclude that DNNs are good models of human object recognition given that humans rely on global shape not texture. Of course, this is just an illustration, and additional forms of second-order confounds may also exist and play a role in driving RSA scores in past research.

It is important to emphasize that confounds are ubiquitous in datasets (Torralba & Efros, 2011) leading DNNs to often classify images on the basis of short-cuts (Geirhos et al., 2020) and it is unclear why confounds would not also drive high RSA scores between DNNs and visual cortex. Our results are also consistent with recent results such as Xu & Vaziri-Pashkam (2021) who reported much reduced RSA scores for a different set of images that included novel objects, and the observation that RSA scores were greatly reduced for "controversial stimuli" (Golan et al., 2020). Similarly, the finding that high RSA scores are obtained for a wide variety of DNNs that differ in fundamental ways (e.g., Conwell et al., 2022; Storrs et al., 2021), suggests that that the RSA scores are driven by features of the image and not specific properties of the DNNs.

We would like to emphasise that our results are not an indictment of RSA per se. Rather, our critique is aimed at the problematic inferences that are frequently being drawn based on RSA when comparing complex systems on uncontrolled and high-dimensional datasets. There are many fruitful ways to use RSA, particularly by constructing theory-based representation dissimilarity matrices and comparing them with observed RDMs. An example of this approach is outlined by Naselaris & Kay (2015), who discuss how RDMs can be constructed based on hypotheses (such as whether

the luminance of an image is driving observed differences between conditions) and compared with the RDMs of observed data. Used in this way, RSA is a useful tool for model-comparison, with different target RDMs corresponding to clearly defined hypotheses. The problem arises when the target RDM is generated by a complex system processing high-dimensional stimuli in which there is no attempt to remove second order confounds that could drive outcomes.

It is important to emphasize that some researchers are manipulating images in theoretically motivated ways to rule out possible confounds when comparing DNNs and brains with RSA (Bracci et al., 2019; Yargholi & de Beeck, 2023; Zeman et al., 2020), but in the overwhelming majority of cases, images are not manipulated to rule out possible confounds. Indeed, there is a strong view in the field that DNNs need to be tested on naturalistic images (Love & Mok, 2023; Summerfield & Thompson, 2023), and there are a growing number of large brain and behavioural datasets composed of naturalistic images of objects and scenes (e.g., Allen et al., 2022; Battleday et al., 2020; Contier et al., 2021; Hebart et al., 2023) that are being used to assess DNN-human alignment with RSA (Conwell et al., 2022; Doerig et al., 2022; Hebart et al., 2023; Mehrer et al., 2021). In these cases, it is difficult to rule out the possibility that second order confounds in images are contributing to high RSA scores.

## 4.2 TWO DIFFERENT VIEWS OF REPRESENTATIONS

One possible approach to sidestep our analysis is to take the view that representational geometry *is* how brains represent knowledge, and therefore, a strong correlation in representational geometries between two systems is sufficient to conclude that DNNs and biological vision are aligned. This question goes to the heart of an existing debate in philosophy, where philosophers distinguish between the *externalist* and *holistic* views on mental representations (Fodor & Lepore, 1992). According to the first view, the content of representations is determined by their relationship to entities in the external world. From this perspective, our finding of high RSA scores between two systems processing different visual features is problematic. Alternatively, a researcher might reject the externalist view and adopt the perspective that representations obtain their meaning based on how they are related to each other within a system, rather than based on their relationship to entities in the external world. That is, "representation *is* the representation of similarities" (Edelman, 1998). From this perspective, as long as the two systems share the same relational distances between internal activations, one can infer that the two systems have similar representations, regardless of the features that drive the geometries. For more discussion of these two philosophical perspectives on representations, see Appendix B.

Two points merit discussion here. First, most neuroscientists and psychologists studying the representations that support biological vision and cognition more generally adopt the externalist perspective. Indeed, the "standard model" of human vision (Riesenhuber & Poggio, 1999) is an extension of the classic Hubel & Wiesel (1959) study that identified simple and complex cells in V1 that represent simple line segments in the world. Indeed, when studying vision, almost all single cell recording studies in neurophysiology and fMRI studies carried out in psychology attempt to characterize how the brain represents visual information in the world, not the relations between internal representations. If researchers comparing DNNs and brains using RSA reject the externalist in favour of the holistic position, then this needs to be made much clearer.

But more importantly, many researchers studying the alignment of DNN and biological vision using RSA are not only making representational claims but also mechanistic ones. That is, claiming a DNN is a good model of biological vision and object recognition is more than a claim that DNNs learn human-like representations, but a claim that the two systems are mechanistically similar. And that entails that the two systems are encoding similar visual features when recognising objects. For instance, it would be odd to claim that a DNN that classifies objects based on texture could be a good model of human object recognition, and indeed, there is now a concerted effort to make DNN classify objects based on shape rather than texture to make the more human-like (e.g., Dehghani et al., 2023).

## 4.3 RELATION TO EXISTING RESEARCH

It is important to note how our results differ from previous studies in neuroscience and computational modelling exploring limitations of RSA. Some of these studies have focused on the importance of

how neural data is pre-processed. For example, Ramírez (2017) found that pre-processing steps, such as centering (de-meaning) activation vectors may lead to incorrect inference about the representational geometry of activations. They demonstrated that subtracting the mean from activations could change the rank order of similarity between conditions. In turn, this could lead to clearly distinct RDMs becoming highly correlated and vice-versa. While this is an important methodological point, it is clearly distinct from the point we are making in this study. Indeed, the results here are agnostic of the data pre-processing steps and hold whether or not activations are centered.

Another set of studies have explored how the procedure of data collection can influence the results of RSA. For example, Henriksson et al. (2015) and Cai et al. (2019) demonstrated that RDMs measured based on fMRI data can be severely biased because of temporal and spatial correlations in neural activity. These authors have pointed out that if activity patterns from different brain regions are recorded during the same trial, the similarity estimates will be exaggerated due to correlated neural fluctuations in these regions. In contrast, the confounds that are highlighted in this study exist in the stimulus itself. Therefore, even if one were to completely mitigate the bias in estimating RDMs, the types of confounds we highlight in our work would still pose problems when drawing inferences from correlation in RDMs. A third set of studies have highlighted the importance of choosing the correct distance metric when using RSA (Ramírez, 2018; Bobadilla-Suarez et al., 2020). Again this is an important issue, but our results demonstrate how different stimulus features can lead to the same representational geometry. This is fundamental to the nature of representational geometries, rather than a consequence of the distance metric used.

Of course, the problem of confounds in stimuli is not unique to RSA. For example, a number of studies have pointed out the problem with confounds in the context of multivariate decoding, where authors have argued that successfully decoding a signal from a neural activation pattern is no guarantee that the signal is encoded by the brain or decoded by downstream processes (Naselaris & Kay, 2015; Weichwald et al., 2015; Ritchie et al., 2019; Hebart & Baker, 2018). In response, researchers have adopted a variety of methods to deal with confounds such as cross-validation (Snoek et al., 2019), confound regression (Kostro et al., 2014), counter-balancing data (Rao et al., 2017) and commonality analysis (Greene et al., 2016). We couldn't agree more with this direction of research and our study highlights two properties of confounds that makes it especially challenging to compare neural representations with those in complex models working on high-dimensional data. Firstly, these confounds are second-order – that is, they are not only category-correlated (as is the case for confounds highlighted for multivariate decoding), but also mimic the second-order similarity structure of the variable of interest. Secondly, when using high-dimensional datasets (such as naturalistic images) and complex target models (such as DNNs) for testing, these confounds are unknown to the experimenter and may be present in the entire dataset. This restricts the utility of existing methods, such as cross-validation and counter-balancing data, for dealing with these confounds, a point made by researchers employing these methods (Rao et al., 2017; Kostro et al., 2014). Indeed, we are unaware of any statistical methods that can completely eliminate confounds under these settings, and instead, experimental designs that test for, and remove confounds are required.

## 4.4 General Recommendations

In closing, we describe our recommendations for practitioners who would like to use RSA for comparing complex systems based on high-dimensional data. First, since the structure of datasets can artificially modulate RSA scores, researchers should compare systems on a wider variety of datasets and sampling schemes than currently done. If high DNN-brain RSAs are observed in some image datasets and not others (as is currently the case), it suggests that there may be some idiosyncratic features of some datasets (e.g., second order confounds) that are driving the higher scores. Second, given that confounding features can lead to similar representational geometries between two systems, researchers should consider running additional controlled experiments that manipulate independent variables designed to test hypotheses to rule out this possibility. This point has recently been made by Bowers et al. (2023) in relation to testing the similarities of DNN and human vision.

Lastly, our most general recommendation is that researchers should highlight more clearly whether they are adopting an externalist or holistic view of representations, and more clearly highlight the possible role of second order confounds in driving RSA scores.

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

Appendix A

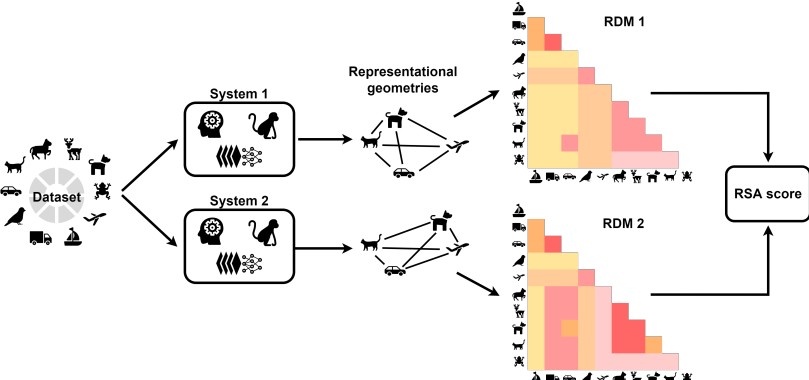

Figure A1: RSA calculation. Stimuli from a set of categories (or conditions) are used as inputs to two different systems (for example, a human brain and a primate brain). Activity from regions of interest is recorded for each stimulus. Pair-wise distances in activity patterns are calculated to get the representational geometry of each system. This representational geometry is expressed as a representational dissimilarity matrix (RDM) for each system. Finally, an RSA score is determined by computing the correlation between the two RDMs. It is up to the resercher to make a number of choices during this process including the choice of distance measure (e.g., 1-Pearson's r, Euclidean distance etc.) and a measure for comparing RDMs (e.g., Pearson's $r$, Spearman's $\rho$, Kendall's $\tau$, etc.).

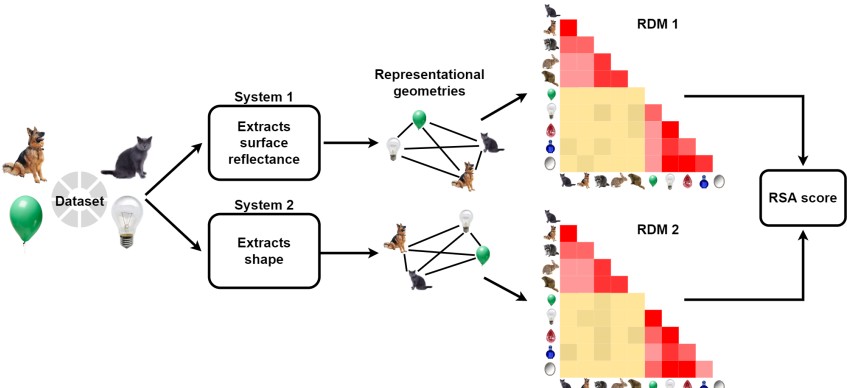

Figure A2: Example of a second-order confound. Two systems, one forming representations based on surface reflectance of objects (while ignoring all other features such as colour or texture) and the other based on global shape (while ignoring other features), can have very similar representational geometries. This similarity would lead to a high RSA score but would not justify an inference about the representations being similar.

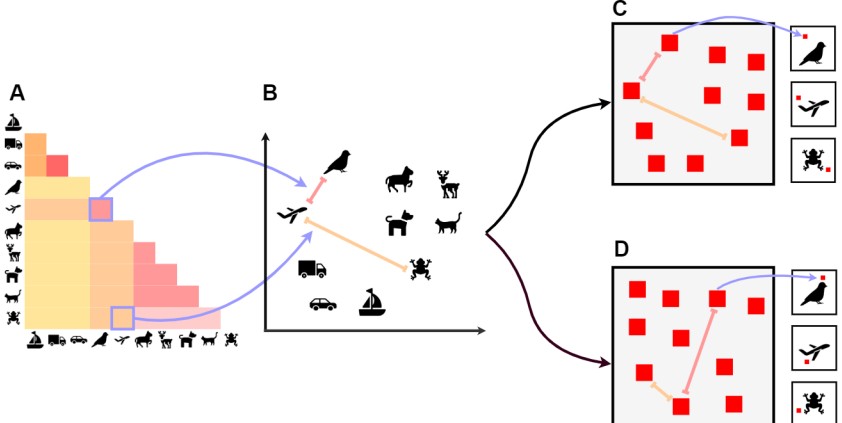

Figure A3: Study 1 confound placement. The representational geometry (Panel A and B) from the network trained on the unperturbed CIFAR-10 images is used to determine the location of the single pixel confound (shown as a red patch here) for each category. In the 'Positive' condition (Panel C), we determined 10 locations in a 2D plane such that the distances between these locations were positively correlated to the representational geometry – illustrated here as the red patches in Panel C being in similar locations to category locations in Panel B. These 10 locations were then used to insert a single diagnostic – i.e., category-dependent – pixel in each image (Insets in Panel C). A similar procedure was also used to generate datasets where the confound was uncorrelated (Panel D) or negatively correlated (not shown here) with the representational geometry of the network.

Appendix B - A brief history of RSA and its philosophical origins

In the 1990s there was an important debate taking place on how to compare the mental representations of two individuals. On one side of this debate was Paul Churchland. Inspired by the success of connectionist models, Churchland argued that the brain represents reality as a pattern of activations over its network of neurons (Churchland, 1989). This pattern of activation can be seen as a position in the brain's (high-dimensional) state-space. So, Churchland argued that one could compare how two individuals represent an object by comparing the corresponding positions in each individual's state-space. On the other side of the debate were Jerry Fodor and Ernie Lepore (Fodor & Lepore, 1996). They pointed out that a problem with Churchland's proposal was that it "offers no robust account of content identity" (p 147). On Churchland's account, they argued, two mental representations have the same meaning only if they are embedded in identical state-spaces. This condition was highly unlikely to be satisfied in practice, given that no two brains have either the same number or connectivity of neurons and no two individuals have exactly the same experiences.

A possible solution to this problem of comparing representations across state-spaces of different dimensions was proposed by Laakso & Cottrell (2000), who were investigating whether different neural networks, trained on the same data, represented an input stimulus in a similar manner. A direct comparison of activations across networks was not possible due to the difference in the number of units. To overcome this problem, Laakso & Cottrell (2000) devised a method that compared encodings based on their *relative* positions in state-space. That is, based on a second-order isomorphism. They argued that two networks could be said to represent a concept in a similar manner if both networks partitioned their activation space (amongst concepts) in a similar manner – that is, if the activation spaces in both systems had a similar *geometry*. Laakso & Cottrell conducted a series of experiments with neural networks, showing that neural networks with different sensory encodings and different number of hidden units nevertheless partitioned their activation space in a similar manner, leading them to conclude that these networks learned similar internal representations.

Churchland (1998) saw Laasko and Cottrell's method as a decisive response to Fodor and Lepore's scepticism. He argued that, using Laasko and Cottrell's method, one could use the state-space approach to compare representations across individuals, even individuals that had different dimensions of their representational spaces. All one needed to do was to replace the requirement of "content identity" with "content similarity". That is, instead of comparing absolute positions of representa-

tions, one could simply compare how representations were organised *relative* to each other within each representational space.

However, Fodor & Lepore (1999) argued that Churchland's reply was, in fact, "an egregious *ignoratio elenchi*" (p. 382). The problem was *not*, they argued, that one couldn't find the right metric to measure similarity across vector spaces of different dimensions. Rather, it was the fact that Churchland (and Laakso & Cottrell, 2000) were interested in a *semantic* similarity – i.e., they wanted to compare whether representations had the same meaning in the two systems. Fodor & Lepore argued that this problem of semantic similarity was intractable because similarity of concepts across systems of different dimensions is undefined. Consider the concept of a 'dog'. Let's say one person's representational space has a dimension of 'loyalty', while the other person's representational space does not. There is no principled answer for how similar the representation of 'dog' should be for these two individuals as it depends on how the dimension of 'loyalty' is weighted in the concept of 'dog'. And the relative weight of dimensions can differ for different concepts and circumstances. Moreover, Fodor & Lepore (1999) argued that even identical representational geometries could *mean* very different things. For example, one individual may represent a dog along the dimensions of 'size' and 'speed' as being small (sized) and medium (speed). Another individual may represent a dog along the dimensions of 'usefulness' and 'furriness' as being of small (usefulness) and medium (furriness). Even if the concept of a dog occupies a similar position in both state-spaces (small, medium) the two individuals clearly represent dogs differently.

Representational Similarity Analysis is an evolution of Laasko and Cottrell's method for comparing representations across systems. It retains its core principle of comparing representations based on their relative locations within each system's state-space. In addition, it formalises the ideas of similarity of representations within and across systems (Kriegeskorte et al., 2008a). Like Laasko and Cottrell's method, a representation is usually coded as a vector of activation over some units (in a neural network or the brain). However, it could also be a behavioural measure, such as similarity judgments or even measures like accuracy or response times. We believe that many of the objections levelled by Fodor and Lepore against Churchland's idea of comparing systems based on relative positions in state-space also hold for representation similarity analysis. For example, Fodor and Lepore's point that similar state-space representations could mean different things can also be extended to RSA and below we will show how different systems with same representational geometries can, in fact, be encoding very different properties of sensory stimuli. If the relation of an activation pattern plays a role in it's meaning, then activations within these systems *mean* very different things and yet have very comparable state-space representations (i.e. geometries). The only way to argue that concepts have a similar meaning in systems with similar representational geometries is to adopt the perspective that the meaning of these activation patterns is determined entirely by their relation to other activation patterns. And as Fodor & Lepore (1999) argued, and we discuss in the General Discussion, adopting this perspective comes with its own set of problems.

