# OpenReview forum: "Inferring DNN-Brain Alignment using Representational Similarity Analyses can be Problematic"
_ICLR.cc/2024/Workshop/Re-Align — ICLR 2024 Workshop Re-Align Poster_

### Official Review · Reviewer_N3QH · 2024-02-20
**An interesting observation about a subtle shortcoming of RSA. Solid Paper.**

**Rating:** 2
**Fit:** 3
**Confidence:** 2

**Workshop Review:**

## Summary
The authors raise a concern about Representational Similarity Analysis (RSA), a popular method that is commonly used in Neuroscience, especially for brain-machine comparisons. They argue that a high RSA score between two systems is not necessarily indicative of their alignment: If two subsets of features exists in the dataset, each of which is sufficient for solving the task, and if these feature sets imply a similar representational geometry, then RSA scores between models using only one of those feature sets will be high, although their strategies for solving the task differ fundamentally. The authors empirically demonstrate the validity of this idea by constructing such datasets, training models on them and observing their high RSA scores in spite of the differences between the features used by the networks.

## Strengths
The work is very relevant to the community and fits the venue well. It raises a valid point that should be a problem in theory and nicely shows that it is indeed possible to construct an example demonstrating this problem in practice. The presentation of the work is clear and easy to follow.

## Weaknesses
- While the work succeeds in proving that there are conditions under which RSA will yield misleading results, it remains doubtful whether such second-order effects really appear in practice. The authors propose texture vs shape as a (very reasonable!) example of potential second-order confounds, but make no attempt at testing whether this is actually the case. Doing so would have made the work outstanding, but it also seems fair to relegate this to further work.
- Section 4, especially 4.2, seems a bit verbose - the paper could be more concise.

## Recommendation
Overall, the paper raises a valid point about how to correctly interpret the output of a commonly used method (specifically, how _not_ to interpret it). In defense of RSA, a requirement for the feasibility of this approach is the similarity of the representational geometries implied by the feature sets - _which is the only thing RSA claims to measure_. But it is a valuable insight that the similarity of the geometry of the representations can be caused more by the stimuli than by the models. We have been discussing the problematic nature of RSA (calculating correlations of correlations…) in our lab for a while now, and I’m happy to see work that makes those concerns concrete. I propose to accept as poster.

## Questions
- The decision to use models pre-trained on ImageNet to then solve CIFAR seems like a slightly odd choice to me, as such models will be overparameterized for the task. Why not train an appropriate, smaller CNN from scratch on CIFAR? This choice should at least be justified. I don’t immediately see how using larger models is a problem, though.
- figure 1 was a bit confusing at first - the blue bars do not correspond to exactly the same datasets, right? For instance, phi-2 to phi-4 were tested on the confounds matching their training distribution, while the blue bar of phi-1 is probably the performance on the un-corrupted test set?
- What models were used for Study 2? You mention that they were taken from torchvision and not pretrained, but which models are they? I’m a bit concerned since there are only 818 training images, so training e.g. a VGG-19 on that training set would just lead to overfitting.
- In figure 3, how would an example for the anticorrelated condition look like? Simply: Points that are close end up far away, and vice versa? Might be more informative to add an image for this condition rather than the random one.

## Additional Feedback
- Didactically, it might be a good idea to define clearly what is meant by a “second-order confound” much earlier in the manuscript. I vaguely got the idea, but it remained imprecise for a few pages.
- Arguably, the comparison of performance (e.g. in Figure 1) does not really show that the network representations are truly different. It just shows that _those parts of the representation that are used by the classifier_ are different - you would technically see the same result for two equal feature extractors with different classifiers stacked on top of them. For example, the model trained on manipulated images (phi-2) could still extract all features of the main model (phi-1) plus the position of the category-pixel. If it then only used the latter feature in its classifier, you would observe the difference in performance on un-manipulated images, while the extracted representations are actually highly similar. It would be better to demonstrate difference of representations not via difference of performance, but by exploiting that phi-1 and phi-2 actually have the same architecture, which enables a more direct comparison of their representations. For example, you could learn an affine transform between their embedding spaces and show that even the optimal transformation performs poorly.
- “most DNNs often classify objects on the basis of texture and local features whereas humans rely on global shape representations” $\rightarrow$ I was surprised that you did not cite Geirhos et al. here.
- “These 5 images were obtained by computing the correlation of average activation elicited by an image with the average activation for the entire category.” $\rightarrow$ Maybe add that this is done to obtain _prototypical_ images for the category?
- I noticed a few typos in the manuscript and point them out for your convenience:
  - “a pixel location diagnostic with object category” $\rightarrow$ “diagnostic of object category”
  - “even if DNNs and humans are aligned in how the represent texture and local features” $\rightarrow$ “how THEY represent”
  - “and it unclear why confounds would not also drive high RSA scores” $\rightarrow$ “it IS unclear”
  - “Our results are also consistent with with recent results such as” $\rightarrow$ “consistent with recent”
  - “it would odd to claim that a DNN” $\rightarrow$ It would BE odd”
  - “our findings are also relevant researchers adopting a holistic theory” $\rightarrow$ “relevant TO researchers”
  - Sometimes, you write DNN instead of DNNs in sentences where it should be plural
  - “Churchland argued that the brain represents reality as a pattern of activations over it’s network of neurons” $\rightarrow$ over its network
  - “Representation Similarity Analysis is an evolution of Laasko and Cottrell’s method” $\rightarrow$ Representational

**Reason For Not Giving Higher Score:**

The paper is limited to an existence proof of datasets for which RSA will "fail", but they are pretty contrived and one could argue that since they are deliberately constructed to have a representational geometry similar to that of the baseline model, RSA is performing as it should.

**Reason For Not Giving Lower Score:**

The paper should be interesting for many practitioners using RSA. It clearly belongs into the workshop.

**Reviewer Domain:**

cognitive science

---

> ### Author Response · Authors · 2024-05-02
>
> Thank you very much for your review and comments.
>
> *Question 1*
>
> Study 1 takes networks pre-trained on ImageNet so that the two sets (normally trained networks and the various networks trained on manipulated datasets) would have the same starting representations. Any manipulated changes would then be exclusively due to tuning on different datasets. Additionally, it showed that even, comparatively, little tuning resulted in a drastic change in representational geometries due to only one pixel difference between the training sets.
>
> *Question 2*
>
> It is correct that performance on perturbed datasets for networks trained on perturbed datasets in Figure 1 is matched to the perturbance they were trained on. E.g., the “positive” networks were tested on a test set in which pixels were placed to positively correlate with the target RDM. These networks perform poorly when tested on a dataset with a different confound (e.g., the “positive” network performs at chance when presented with images that have a pixel placed to negatively correlate with the target RDM). We have added a clarification to the figure caption. Phi-1 (“normal”) was tested on all datasets.
>
>  *Question 3*
>
> We have now clarified that VGG-16 networks were used in Study 2 as well. We agree that the dataset is quite limited – but it is from an important publication. Additionally, training is short at 10 epochs and test results indicate solid performance without overfitting.
>
>  *Question 4*
>
> Figure 2 shows an example of a hierarchical (positive) placement. We do not show what an anti-correlated placement would look like as we do not have that condition in Study 2, but the logic in the comment is correct. Kinds that one would expect to be close in a representational geometry (e.g., human faces and animal faces) would have confounds placed at large distances. On the other hand, kinds that one would expect to be distant in representational geometries would have confounds placed in close vicinity (e.g., man-made objects and animal bodies).

---

### Official Review · Reviewer_cE5K · 2024-02-24
**A strong case for reconsidering the use of RSA in comparing DNN and brain representations**

**Rating:** 3
**Fit:** 3
**Confidence:** 3

**Workshop Review:**

Summary: The authors critically examine the use of RSA in inferring alignments between DNNs and human brain activations in the context of visual object recognition. The authors highlight one significant issue: high RSA scores can be achieved between systems employing distinct object classification strategies due to second-order confounds in image datasets. They further argue that such confounds are likely present in current and past research datasets, suggesting that RSA might not be a reliable tool for studying DNN-human alignment without careful experimental manipulation of images to remove these confounds. The implications of this work urge reconsideration of conclusions drawn from RSA studies and the adoption of more rigorous experimental designs.

**Reason For Not Giving Higher Score:**

Some questions for the authors and points of feedback:
1. How generalizable do you believe the simulation studies' findings are across different datasets and DNN architectures?
2. Given the critique of RSA for aligning representations, what alternative methodologies would you recommend?
3. The authors assume that single-pixel confounds in the modified CIFAR-10 datasets can simulate second-order confounds in real-world datasets. Simplifying single-pixel modifications might not capture the multi-dimensional and interactive nature of real-world confounds, limiting the generalizability of the findings.

Also relevant work in the field to include in the citations are:
1. Diedrichsen, J., & Kriegeskorte, N. (2017). Representational models: A common framework for understanding encoding, pattern-component, and representational-similarity analysis. PLoS computational biology, 13(4), e1005508.
2. Xu, Y., & Vaziri-Pashkam, M. (2021). Limits to visual representational correspondence between convolutional neural networks and the human brain. Nature Communications, 12(1), 2065.
2. Nagaraj, A., Ashok, A. K., Linsley, D., Lewis, F. E., Zhou, P., & Serre, T. (2023). Ecological data and objectives align deep neural network representations with humans. In UniReps: The First Workshop on Unifying Representations in Neural Models.
3/ Kriegeskorte, N., Mur, M., & Bandettini, P. (2008). Representational similarity analysis - connecting the branches of systems neuroscience. Frontiers in systems neuroscience, 2, 4.
4. Geirhos, R., Jacobsen, J.-H., Michaelis, C., Zemel, R., Brendel, W., Bethge, M., & Wichmann, F. A. (2020). Shortcut learning in deep neural networks. Nature Machine Intelligence, 2(11), 665–673.

**Reason For Not Giving Lower Score:**

1. The paper empirically demonstrates that high RSA scores can be achieved between systems that classify objects using qualitatively different features due to second-order confounds in image datasets​​. This finding is critical as it challenges the assumption that high RSA scores indicate a meaningful similarity in representational geometries between humans and DNNs.
2. The results of this paper are highly relevant to the ongoing debate regarding the relationship between DNNs and biological vision​​. The paper contributes significantly to the discourse on how we should interpret similarities in representational geometries between artificial and biological systems.
3. The paper's findings have significant implications for future research, urging a reevaluation of past RSA studies and the adoption of more nuanced methodologies that account for potential confounds​​.

**Reviewer Domain:**

machine learning

---

> ### Author Response · Authors · 2024-05-02
>
> Thank you for your insight and comments.
>
> Question 1 and 3 tackle a very similar issue of how well these results generalize, or how prevalent second order confounds are in other datasets / how susceptible other architectures are at exploiting them.
>
> These are important questions which are difficult to answer. It would amount to proving a confound influenced alignment in a given dataset and what that confound is. It is precisely because of the complexity of real-world images and the network architectures that this is difficult to assess.
>
> There is no telling what combination of features may form a confound. Some good experimental work has shown convolutional neural networks certainly learn to recognize objects based on texture. However, texture, colour, shape - these features are simple to conceptualize and understand to humans - while deep neural networks may be utilizing some combination of information within stimuli which we would never think of as being a feature.
>
> Similar biases (which are dissimilar) to humans are better researched via experimentation than measuring alignment. This answers the second question as well, careful experimentation is the primary approach we advocate in order to determine how networks represent inputs and whether they do so in a manner similar to humans.

---

### Official Review · Reviewer_hsUo · 2024-02-26
**Excellent work demonstrating the flaws in the conclusions drawn from typical RSA analysis**

**Rating:** 3
**Fit:** 3
**Confidence:** 3

**Workshop Review:**

Overall this is an excellent paper that I believe will be of interest to the community so I recommended it for a talk. RSA is a widely used technique and although its limitations are known to many the conclusions drawn from computing a similarity of similarities are often ignore these limitations. I agree with the authors statement "We hope our simulations motivate researchers to reexamine the conclusions they draw from past research and focus more on RSA studies that manipulate images in theoretically motivated ways"

Clarity:
This paper is clear and easy to follow. The experiments were explained well and there was a thoughtful discussion.

Novelty/interest to the community:
This paper is a good topical fit for the workshop and I believe many in the community will be interested in the experiments and discussion. While not entirely novel, the experiments are well thought out and provide a specific examples of flaws with using RSA in 2 popular settings CNN/CNN and CNN/Brain. I have not seen specific examples of these limitations in context explained well by a simulation before.

Correctness:
I believe the experiments and results in this paper are correct.

**Reason For Not Giving Higher Score:**

N/A

**Reason For Not Giving Lower Score:**

This is an important topic as RSA is used in many papers by members of this community. The paper is well written and will hopefully spark interesting discussions about the validity of the conclusions draw by prior work that relies on RSA.

**Reviewer Domain:**

machine learning

---

### Decision · Program_Chairs · 2024-03-02

Accept (Poster)